# Laponite Composites: In Situ Films Forming as a Possible Healing Agent

**DOI:** 10.3390/pharmaceutics15061634

**Published:** 2023-05-31

**Authors:** Ramón Andrés Pineda-Álvarez, Carolina Flores-Avila, Luis Medina-Torres, Jesús Gracia-Mora, José Juan Escobar-Chávez, Gerardo Leyva-Gómez, Mohammad-Ali Shahbazi, María Josefa Bernad-Bernad

**Affiliations:** 1Facultad de Química, Universidad Nacional Autónoma de México, Ciudad Universitaria, Cto. Exterior S/N, Coyoacán, Ciudad de México 04510, Mexico; and_res5@comunidad.unam.mx (R.A.P.-Á.);; 2Facultad de Estudios Superiores Cuautitlán, Unidad de Investigación Multidisciplinaria-L12 (Sistemas Transdérmicos), Universidad Nacional Autónoma de México, Carretera Cuautitlán-Teoloyucan, km 2.5 San Sebastián Xhala, Cuautitlán Izcalli 54714, Mexico; 3Department of Biomedical Engineering, University Medical Center Groningen, University of Groningen, Antonius Deusinglaan 1, 9713 AV Groningen, The Netherlands; m.a.shahbazi@umcg.nl; 4W.J. Kolff Institute for Biomedical Engineering and Materials Science, University of Groningen, Antonius Deusinglaan 1, 9713 AV Groningen, The Netherlands

**Keywords:** laponite, maltodextrin, sodium ascorbate, wound dressing, films, composite

## Abstract

A healing material must have desirable characteristics such as maintaining a physiological environment, protective barrier-forming abilities, exudate absorption, easy handling, and non-toxicity. Laponite is a synthetic clay with properties such as swelling, physical crosslinking, rheological stability, and drug entrapment, making it an interesting alternative for developing new dressings. This study evaluated its performance in lecithin/gelatin composites (LGL) as well as with the addition of maltodextrin/sodium ascorbate mixture (LGL MAS). These materials were applied as nanoparticles, dispersed, and prepared by using the gelatin desolvation method—eventually being turned into films via the solvent-casting method. Both types of composites were also studied as dispersions and films. Dynamic Light Scattering (DLS) and rheological techniques were used to characterize the dispersions, while the films’ mechanical properties and drug release were determined. Laponite in an amount of 8.8 mg developed the optimal composites, reducing the particulate size and avoiding the agglomeration by its physical crosslinker and amphoteric properties. On the films, it enhanced the swelling and provided stability below 50 °C. Moreover, the study of drug release in maltodextrin and sodium ascorbate from LGL MAS was fitted to first-order and Korsmeyer–Peppas models, respectively. The aforementioned systems represent an interesting, innovative, and promising alternative in the field of healing materials.

## 1. Introduction

A wound is defined as an epithelial rupture of the skin that is classified as acute or chronic. Within them, they are subdivided into pressure ulcers, venous leg ulcers, and diabetic foot ulcers, as well as traumatic and surgical wounds. These lesions increase the risk of infections, while internal tissues are exposed to damaging factors [1]. For this reason, in clinical fields, the closure and repair of wounds after a traumatic event or surgery are vital. Proper treatment will result in increased survival of patients through adequate healing [2]. However, the wound healing process requires many processes at the cellular level, some of which remain unknown. 

Using nanotechnology through nanocarrier systems that facilitate the administration of drugs and cover wounds to protect them could be an alternative to meet the ideal conditions of healing materials such as laponite composites. Laponite is a mineral clay that is widely used as an excipient and active substance in the pharmaceutical and cosmetic industry. Laponite composites are described as disks with an approximate diameter of 25 nm and a width of approximately 1 nm. This clay has a negative charge on the inner surface and, at the edges, is weakly positive [3,4]. Laponite’s external sheets are tetrahedral and octahedral in the middle (2:1)—where Mg^2+^ and Li^+^ cations are found. The weaker positive charges interact with the opposing surfaces of the adjacent particles when suspended in water. The process can go on to give a “house of cards” structure, causing a highly thixotropic gel, which is referred to by researchers as physical crosslink [5]. This clay is attractive for regenerative medicine and tissue engineering [6,7] due to its degradation in a physiological environment; for wound healing, the most important are Mg^2+^ and Na^+^. Magnesium is an essential element in the human body, as it is involved in producing all polyphosphate compounds in cells (adenosine triphosphate, ATP), enzyme activity and ion channel action, metabolic pathways, and other processes. Sodium plays a role in maintaining electrolyte and fluid balance [8]. Additionally, the gels of this clay have previously been used for wound dressing and drug delivery purposes to carry amino acids such as arginine, lysine, and leucine to wounds, promoting human skin fibroblast proliferation [9]. Ghandiri and coworkers described a developed laponite/alginate/mafenide nanocomposite studied in vitro. In addition to its antibacterial activity, it was able to absorb exudate from wounds where it was administered due to the swelling property of the laponite. Moreover, the Mg^2+^ ions in its chemical structure helped to reduce the cytotoxicity of mafenide in cellular fibroblasts, which are in charge of collagena synthesis in the proliferation phase of the healing process [10]. Another function of this clay is that its structure facilitates physical crosslink via interactions between positive and negative charges, and this was confirmed by Golafshan in a PVA/Alginate network, who explained how this feature depends on its concentration. At the same time, laponite dramatically influenced the mechanical properties of the materials [11].

Teng et al. (2022) and Rajabi et al. (2020) combined gelatin with laponite to develop hydrogels with hemocompatibility properties. The hybrids were described as having good compatibility and showed a hemostatic function because the negative surface of the clay attracted blood components and accelerated the clotting process. They suggested their materials could be applied to wounds to avoid hemorrhages [12,13]. On the other hand, in clinical studies, the topical application of maltodextrin/sodium ascorbate in wounds has shown to be a protective cover for the invasion of microorganisms, they also promote the stability of the humidity of the medium for tissue granulation and have bacteriostatic activity due to pH reduction. Due to gradual metabolism, maltodextrin can release glucose into the deep wound environment, providing topical nutrition to the tissue. In vitro studies showed the chemotactic effect of this sugar on polymorphonuclear leukocytes, benefitting healing. The action mechanism followed by maltodextrin/sodium ascorbate to accelerate the healing process relies on whether its conjunction with transforming growth factors (TGF) at the fibroblast level cause the rapid expression of a broad spectrum of enzymes related to collagena rechange (collagenases and gelatinases), accelerating the proliferation phase [14].

This study evaluated the performance of laponite in lecithin/gelatin composites (LGL) vs. loading them with a maltodextrin/sodium ascorbate mixture (LGL MAS). The innovative pharmaceutical forms were applied in a liquid state (nanoparticle dispersion) to increase contact with irregular surfaces, from which films were formed as possible dressing materials. Additionally, this research proved that the nanocarrier function of the clay nanodisks is to deliver maltodextrin and sodium ascorbate. These experiments offer insights into changes in the physicochemical properties of the particles and films developed with lecithin/gelatin/laponite and explain the benefits and limitations of adding this clay into them.

## 2. Materials and Methods

### 2.1. Materials

Laponite EP (LAP) and sodium ascorbate (AS) were donated by BYK Additives & Instruments and AVIMEX (CDMX, Cuautitlan Izcalli, Mexico), respectively. Maltodextrin DE 16.5–19.5 (M), gelatin from bovine skin type B strength ~225 g bloom (GEL), L-α-Phosphatidylcholine from soybean type IV-S, ≥30% (enzymatic) (LEC), glutaraldehyde grade I 25% (*v*/*v*) were purchased from Merck, while acetone and potassium chloride were purchased from J. T. Baker, and other analytical-grade chemicals and solvents were purchased from Sigma-Aldrich (St. Louis, MO, USA).

### 2.2. Preparation of Nanoparticles and Films

The nanoparticles were prepared by the desolvation method previously reported [15], from 3 dispersions (see Figure 1): (a) 100.0 mg of gelatin dispersed in 3 mL of deionized water at 45 °C, then 3 mL of acetone was added as a desolvating agent. The precipitate was redispersed in the same water volume at 45 °C, and the pH was adjusted to 3 with 0.2 N HCl. (b) For formulations only with laponite, the dispersion contained 8.8 mg of clay in 3 mL of deionized water, while others with maltodextrin/sodium ascorbate mix (M:AS) were supplemented with 10 mg of each drug, and (c) 125.0 mg of lecithin was dispersed in 3 mL of acetone: methanol 1:1. Each dispersion was added consecutively as they were prepared under 18,000 rpm for 2 min using T-18 ULTRA-TURRAX^®^ (IKA). Acetone: methanol was allowed to evaporate for about 24 h under constant stirring (1000 rpm), and subsequently, the pH was adjusted to 6.5 with 0.2 N NaOH. Finally, 100 μL of glutaraldehyde 2% (*v*/*v*) was added as a crosslinking agent. The resulting dispersion was stirred for 24 h more.

The films were prepared by solvent evaporation. A total of 4 mL of dispersed nanoparticles was put in a square mold with an area of 9 cm^2^ for 30 min at 25 °C.

### 2.3. Characterization of Nanoparticles

#### 2.3.1. Particle Size, Polidispersity Index (PDI), and Zeta Potential (ξ)

Dynamic Light Scattering technique (DLS) was applied by using a Zeta-sizer Zen 3600 (Malvern Instruments, Malvern, UK) with a detection angle of 173° at 25 °C ± 1 °C. For each measurement, 50 μL of sample plus 1 mL of deionized water were added to a 1 cm glass disposable cell. For the Zeta potential, the cell was changed to a disposable double capillary cell, maintaining the same angle, temperature, and dilution, and deionized water was added. All measurements were taken in triplicate under room temperature appear in mean ± SD.

#### 2.3.2. Entrapment Efficiency

*Maltodextrin* (*M*). The phenol-sulfuric acid method measured the non-entrapment maltodextrin (see Figure 2a) [16]. Using an Amicon Ultra 0.5 mL Ultracel 10K filter in Eppendorf tubes, 0.5 mL of nanoparticle dispersion was centrifuged at 15,000 rpm for 30 min. The filtering was diluted with 100 μL in 2 mL of deionized water. A total of 50 μL of diluted dispersion was extracted and put in an assay tube, then 50 μL of deionized water, 200 μL of phenol 6.5% (*p*/*v*), and 550 μL of sulfuric acid concentrated were added consecutively. The samples were stored at 4 °C for 30 min and analyzed using the ultraviolet–visible (UV–VIS) spectrometry method at 491 nm (S2000 Ocean Optics Inc. Spectrophotometer, DT-1000CE-BT Tungsten source, SAD500 Ocean Optics Inc. Serial Port Interface, and 1 cm quartz cuvette). The analyzed drug corresponded to the non-entrapment drug. The percentage of entrapment efficiency was calculated according to the following Equation (1):(1)$$\%\;\mathrm{Entrapment}\;\mathrm{efficiency}=\frac{{\mathrm{mg}\;\mathrm{AS}}_{\mathrm{used}\;\mathrm{in}\;\mathrm{formulation}}-{\mathrm{mg}\;\mathrm{AS}}_{\mathrm{analyzed}}}{{\mathrm{mg}\;\mathrm{AS}}_{\mathrm{used}\;\mathrm{in}\;\mathrm{formulation}}}\times 100$$

*Sodium ascorbate* (*AS*). The diluted dispersion prepared before was analyzed via ultraviolet–visible (UV–VIS) spectrometry at 267 nm (see Figure 2b). To calculate the percentage of entrapment efficiency, the same equation used to calculate that of maltodextrin was used. 

Figure 2 shows the calibration curves for each drug. Maltodextrin was analyzed from 42 to 210 μg/mL, and sodium ascorbate from 2 to 10 μg/mL. All measurements were taken in triplicate.

#### 2.3.3. Rheology Study

The samples were characterized in a stress-controlled model TA Instruments Discovery HR3^®^ rheometer (TA Instruments, New Castle, DE, USA) with a concentric cylinders geometry (21.96 mm outer diameter, 20.38 mm inner diameter, 59.9 mm height and 500 μm gap). Viscosity measurements under steady-state simple shear flow ranged from 1 to 300 s^−1^ at 30.5 °C (skin temperature [17]). Viscoelastic properties (i.e., storage and loss modulus) under small-amplitude linear oscillatory flow (i.e., γ < 5%) ranged from 1 to 300 rad/s. The study of the stability of the fluid concerning temperature was carried out in a temperature range of 18 °C to 50 °C (the temperature in which the skin can be found, depending on the stimuli of the environment [17,18]).

#### 2.3.4. Morphology

The optimal formulations were observed via Transmission Electron Microscopy (TEM), which was performed by using a JEM-2010 microscope (JEOL Inc., Peabody, MA, USA). The dilution consisted of 5 μL of sample plus 1 mL of deionized water.

### 2.4. Characterization of Films

#### 2.4.1. Film Thickness

Through the use of a Digital Micrometer CHESC05001 (CHE SCIENTIFIC, resolution of 0.001 mm), we randomly measured the films’ thickness in four positions, and the mean values were used in the calculations.

#### 2.4.2. Mechanical Properties

Each film (3 × 3 cm) was placed between the two jaws of the Texture Analyzer instrument XT plus (TA instruments) at a constant crosshead speed of 50 mm/min with a load cell of 50 N to determine Young’s modulus and tensile strength (three samples of each formulation).

#### 2.4.3. Swelling Behavior and Mass Loss by Solubilization

The degree of swelling in the films was determined by weighing the films before (*M_o_*) and after (*M_f_*) immersion in 20 mL of deionized water for 15 min at 25 °C. Excess water was removed via the use of filter paper. The following Equation (2) calculated this response:(2)$$\%\;\mathrm{Degree}\;\mathrm{of}\;\mathrm{swelling}=\frac{M_{f}-M_{o}}{M_{o}}\times 100$$

For loss by solubilization, the samples were stored in a desiccator with silica gel at 25 °C for 12 h. Each film was weighed (*M_f_*, in this case), and the results were reported as absolute values using the last equation.

#### 2.4.4. Occlusive Effects

The occlusive properties of composite films were evaluated using De Vriger’s in vitro modified method [19]. In this test, ≈100 mg of NaOH was put in a vial, covered with the film, and weighed. The hygroscopic basis captured water vapor that had entered through the sample. This response was calculated by comparing the differential weight loss (%) from when the trial started and when it finished (after 24 h). As a positive control, 100% of water vapor absorbed by the basis was transferred into a vial with NaOH without cover and was related to each measurement.

#### 2.4.5. Clarity

Transmittance (%T) at 620 nm in a spectrophotometer (S2000 Ocean Optics Inc. Spectrophotometer, DT-1000CE-BT Tungsten source, SAD500 Ocean Optics Inc. (Orlando, FL, USA) Serial Port Interface, and 1 cm quartz cuvette) determined the clarity of the optimal formulations.

#### 2.4.6. Bioadhesion and Postwetting Bioadhesion

Penetrometer I173XX (Agrosta, Normandy, France) configured in texture profile analysis mode measured the responses of the three samples analyzed (*n* = 3). The instrument compressed the optimal film (which was 1 cm in diameter) against the pig skin twice at 10 mm/s with an activation charge of 5 gf and a return speed of 5 mm/s, and paused for 5 s between the cycles. For post-wetting bioadhesion, the skin area was moisturized 10 min before the test.

#### 2.4.7. ATR-FT-IR

The attenuated total reflectance–Fourier-transform infrared (FT-IR) spectra were recorded using a UATR Two spectrophotometer (PerkinElmer, Waltham, MA, USA) from 500 to 4000 cm^−1^ with a spectral resolution of 2 cm^−1^. 

#### 2.4.8. Differential Scanning Calorimetry (DSC)

A DSC 1/700 (Mettler-Toledo, Columbus, OH, USA) was used to measure the film’s glass transition temperature (Tg). Calibration employed an Indium standard (melting temperature: 156.6 °C). Approximately 5 mg film was weighed directly into standard aluminum pans. Thermal events were recorded in a temperature range from 8 to 225 °C with a 5 °C/min ramp under a nitrogen atmosphere (purge at 20 mL/min).

#### 2.4.9. Surface Morphology—Scanning Electron Microscopy (SEM) and Atomic Force Microscopy (AFM)

Via the use of SEM JSM-5900 (JEOL USA, Inc., Peabody, MA, USA), the optimal films deposited on the holder coated with gold were observed, obtaining a conductive surface in vacuum conditions. 

Finally, the same films were analyzed using an Atomic Force Microscope NX10 (PARK) equipped with software for image analysis XEI (4.3.4. Build 22, PARK Systems Corp., Suwon, Republic of Korea), using the Non-Contact Mode with a resonance frequency of 230 kHz and a scan speed of 1.0 Hz. The treatment of the images was performed using 5 × 5 μm of the sample.

#### 2.4.10. Uniformity of Content

Each optimal film was left in 3 mL of dimethyl sulfoxide (DMSO) for 3 h (ten samples analyzed, *n* = 10), with the addition of 7 mL of deionized water, and then 1 mL of the new dispersion was centrifuged at 15,000 rpm for 30 min. For each drug, the supernatant was diluted at 100 μL in 2 mL of deionized water and subsequently analyzed using the validated UV–VIS spectrometry method described previously.

#### 2.4.11. Drug Release

Franz Cells were used to evaluate the release of the drugs, using regenerate cellulose dialysis bags MWCS 12–14 kDa (Spectra/Por, Los Angeles, CA, USA) as membranes separating donor and receptor compartments. Optimal composite films that were 1 cm in diameter and supplemented with the drugs (LGL MAS) were evaluated (six samples analyzed, *n* = 6). The dissolution medium was added to 3.5 mL of phosphate-buffered solution (PBS) with pH = 7.4 at 37.5 °C and 50 rpm. A total of 250 μL PBS was extracted at different times for drug quantifications by applying the previously mentioned spectrophotometric methods. The data were fitted to various kinetic models, such as zero-order, first-order, Higuchi, and the Korsmeyer–Peppas model. The model with the highest correlation coefficient was considered the best fit.

*Statistical analyses.* The experimental data were analyzed by using STATGRAPHICS Centurion XVI^®^ software (version 16.2.04, StatPoint Technologies, Inc., Warrenton, VA, USA).

## 3. Results and Discussion

### 3.1. Preparation of Nanoparticles

The excipient charges were the principal property taken into consideration for the manufacture of composites. The desolvation method consisted of three dispersions, as described in Figure 1. In dispersion A, the addition of a desolvating agent reduced the water available to keep the gelatin dispersed, resulting in the contraction of polymer chains; the hydration became too low for larger chains and then precipitated (desolvation 1). This phenomenon improved the distribution and reproducibility of the final product. Then, pH was adjusted to 3—below gelatin’s isoelectric point (pI ≈ 4–5)—to charge it positively [15]. Dispersion B consisted of amphoteric laponite for LGL formulations, while for LGL MAS, it contained the clay and maltodextrin/sodium ascorbate mixture. In the latter, the positive edges of the solid disk formed electrostatic interactions with ascorbate anions. In the case of maltodextrin (a neutral carbohydrate), it developed physical interaction mainly with the negative surface of the mineral [8,20]. Finally, dispersion C had a negative charge due to the presence of soy lecithin (phosphatidylcholine). During the mixing process with the rest of the dispersions, the acetone/methanol mix exhibited desolvating activity upon gelatin, contracted the chains of protein (desolvation 2), and formed the particles by means of violent agitation. At the end of this process, the pH of the dispersion was around 3–4. This condition increased the hydrophobicity of the lecithin and caused particles aggregation [21] while having a possible negative effect upon wounds, if the composites fitted as dressings [22]. For those reasons, pH was adjusted to 6.5, and glutaraldehyde was added as a covalent crosslinker of the free amino groups of the gelatin chains through a Maillard reaction [15]. The films were formed by water evaporation, allowing the coalescence of the particles [23], and the clay acted as a physical crosslinker [24], helping to accelerate the drying process. This research aimed to shed light on the properties of the laponite-based film used on wounds in a liquid state, which will help to increase contact on the irregular surface of wounds.

### 3.2. Characterization of Nanoparticles

#### 3.2.1. Particle Size, Polydispersity Index (PDI), and Zeta Potential (ξ)

*Particle size.* When designing the composite dressing, particle size played an important role. Recently, it has been reported that a particle size of ≤473 nm accelerates the passage from the inflammation to the proliferation phase in the healing process, while particles with a size of ≥1 μm are only able to reduce feverish states and inflammation in the affected area, although their mechanisms of action are unknown [25,26,27]. In this research, the nanoparticles with 8.8 mg of laponite, regardless of the condition evaluated, exhibited sizes ≤ 473 nm (see Table 1). This assay demonstrated that the particulate size decreased when the clay concentration increased. This is related to Laponite’s physical crosslinker effect, which attracted the excipients into the center of the particle [28], thus aiding in the general crosslinking process. Furthermore, the emulsification activity of lecithin showed the same tendency as previously reported (the higher the concentration, the more compact the particles were [29]).

*Polydispersity index (PDI).* The successful formulation of safe, stable, and efficient particles requires the preparation of homogeneous populations of a certain size; hence, a value of PDI ≤ 0.3 is recommended [30]. The results for this response did not show a trend (see Table 1) due to the structural heterogeneity [31]. Nevertheless, the formulations with L:G mix 125:100 and 8.8 mg of laponite had the best performance.

*Zeta Potential determinations (ξ).* The necessary repulsion charge to avoid the aggregation of the particles stands over ± 30 mV [15]. It has been reported that particulate systems have better stability; hence, the interior and exterior formation of the house of cards structure is related to the presence of a clay [32]. Clay concentration plays a significant role upon dispersion stability given the fact that, in high concentration, the neighboring particles are highly attracted by charges, causing agglomerations and leading to precipitation. On the other hand, when not in sufficient concentration, precipitation occurs as a result of poor repulsion between them; thus, the challenge is to find a sufficient balance between laponite and excipients in order to maintain dispersion stability [28,33]. Although there is a direct correlation between laponite concentration and the increased of the negativity charge of particles, the blank samples reflected the most negative results, demonstrating a significant electrostatic interaction between the excipients and the clay [31,34]. The formulations with 150:100 mg:mg L:G mix level precipitated after two weeks—in accordance with their results with values close to zero. The lack of repulsion interaction in these formulations derives from the fact that the ratio between laponite and the rest of the formulation is considerably low, thereby making it difficult to form and preserve the house of cards structure and consequently causing precipitation (see Table 1).

#### 3.2.2. Entrapment Efficiency

Table 1 shows how laponite significantly increased this response in comparison to the blanks for both drugs, although, at first glance, results appeared to be rather equal regardless of the laponite concentration. To determine the significant differences between samples, Fisher’s least significant difference (LSD) procedure was used. Through utilizing this procedure, it is possible to state that were no statistically significant differences between the levels containing 4.4 and 6.6 mg of laponite for both L:G mix levels. This assay demonstrated that the 8.8 mg-containing clay level was the highest encapsulation of M and AS. Moreover, this assay proved the specificity of the excipients to ascorbic anions due to their hydroscopic affinity and charge [35,36].

Based on the results, the particles with the 125:100 mg:mg L:G mix and 8.8 mg laponite, with and without M and AS, were taken as the optimal formulation (LGL and LGL MAS, bold in Table 1). Nevertheless, the complete level of L:G mix in the selected formulation was further studied in films in order to obtain insights about the influence of laponite in regard to mechanical properties and other factors relevant to their possible use as wound dressing materials.

#### 3.2.3. Rheological Analysis

The purpose of evaluating the rheological properties of the samples was to determine their response to flow and gain a better knowledge of the type of fluid to which they correspond at 30.5 °C (mean skin temperature) and their stability at temperatures from 18 °C to 50 °C. This was mainly due to the fact that the skin temperature depends on the surrounding environment, leading to fluctuations in its values in the range evaluated [17,18]. The results presented here are further complemented by the following section concerning Differential Scanning Calorimetry (DSC).

While assessing the viscosity of the optimal formulations vs. shear rate [37], it was observed that both dispersions follow a non-Newtonian behavior, specifically for thinning fluids (*n* < 1), as shown in Figure 3a. This means that, at higher shear rates, the fluids decreased their viscosity. For LGL fluid, the initial values were low (slight slope, *n* << 1) in comparison to LGL MAS. The observed trend was related to smaller particle size (286.4 ± 28.7 nm) and distribution regarding LGL, as well as its higher repulsion charge (−34.5 ± 1.7 mV). In contrast, the other formulation was characterized by a bigger size and a Z potential closer to zero, which probably meant that some particles agglomerated, creating a resistance to flow and therefore obtaining higher values of viscosity. Figure 3b displays the small-amplitude oscillatory flow results [38], evaluating the elastic and viscous modulus (G′ and G″, respectively) vs. frequency-flow (ω). Both optimal fluids demonstrated the same viscoelastic behavior; however, the elastic modulus predominated at all evaluated frequencies (G′ > G″) mainly due to the entanglement between biopolymers and clay in the dispersions [39]. Additionally, the M:AS mix did not affect the evaluated viscoelastic properties.

Finally, the thermal stability test of the optimal formulations for both LGL and LGL MAS at a temperature range from 18 to 50 °C is described in Figure 3c (demarcated by pink and violet curves, respectively), wherein the elastic modulus dominated over the viscous modulus (G′ > G″), meaning that the gel state remained in this temperature range. To gain a better understanding of the effects of laponite on the samples, a non-optimal formulation was analyzed as well. This non-optimal formulation contained 6.6 mg of laponite (LGL 6.6, green). It was observed that when the temperature increased above 31 °C, its elastic modulus decreased (G’), implying that a change had occurred in the solid component of this formulation due to the gelatin (one of the most abundant excipients). This biopolymer is sensitive to temperature and flow since it completely loses its solid form and turns into a weak gel state at 31 °C. The material was not homogeneous and needed more laponite (clay) to avoid this phenomenon, as shown by the optimal formulations [40].

#### 3.2.4. Morphology

Figure 4 shows the development of the particles. TEM confirmed the results obtained via DLS. Figure 4b,d,f,h correspond to the zoom on the edge of each type of particle in the images. The aggregation of some clay platelets at the edge of the optimal particles (see Figure 4f for LGL and Figure 4h for LGL MAS) was observed as described by Negrete and coworkers in their research [41]. The clay can be observed as black lines that are ∼1 nm thick and 25 nm length. The blank vs. laponite particles with M:AS (see Figure 4c,d,g,h) are completely different from one another. The first one, Figure 4c, had a spherical form, where excipients occupied the whole particle, as seen in the close-up presented in Figure 4d. The polymers observed were similar to tiny particles, given the natural incompatibility that prevails between protein-lipid-carbohydrate systems [42], while, in LGL MAS, the excipients were located in the center of the material with a heterogeneous form, being mainly capsular.

### 3.3. Characterization of Films

#### 3.3.1. Film Thickness

Table 2 reveals a statistically significant difference (ANOVA, *p* < 0.05) in which the clay amount and drug mix affected this response. Whereas, laponite concentrations, as well as the addition of M:AS, were proportional to thickness. Other studied features of the films have a strong relationship with this response, as will be discussed in the following paragraphs.

#### 3.3.2. Mechanical Properties

The mechanical properties of a film vary depending on the polymers (structure and degree of polymerization), solvent, pH, plasticizer, and manufacturing process [43]. Thus, when the Texture Analyzer obtains the force-displacement plot, the Tensile Strength (TS) and the Young’s modulus (elastic modulus) can be measured. TS measures the film strength as fracture force experimented by the transverse section area of the material, while the elastic modulus expresses the material’s resistance to deformation. In order to resist the natural deformations of the human skin, the TS of films should be in the range of 2.5–16 N/mm^2^ (MPa) and have closer Young’s modulus values to that of the skin tissue, which can hold values from 4.6 to 20 MPa [44].

Table 2 shows how laponite decreased the stiffness of the films. When comparing formulations with and without M:AS, it can be observed that the addition of the drug mixture improves the elastic module. As previously mentioned, high values of this property are not suitable for materials intended for use as wound dressing materials. Regarding the TS, an opposite effect can be observed, and this response increased with clay because a high laponite content would lead to a highly crosslinked film network, making materials more resistible [28]. Only the optimal formulations had values that corresponded to those selected as specifications. In comparison to LGL, LGL MAS had higher amounts due to the presence of the polysaccharide, as explained by Castro and coworkers. Maltodextrin DE17–19 on hydrogels for films (carbohydrate used in this study) had a superior elevated plastic character and less-entangled polymer chains, making the material more resistant [45].

#### 3.3.3. Swelling Behavior and Mass Loss by Solubilization

For studies such as the one described in this paper, swelling expresses the degree to which the film can absorb water, PBS, or other substances that simulate the exudate, thus allowing the release of the incorporated drug. In relation to this, excessive swelling can cause a reduction in the film’s integrity due to the formation of a free-flowing gel, and this is known as loss by solubilization [46]. The blanks were impossible to analyze because they dispersed on water, revealing that the particles were independent and needed another polymer or crosslinker to stay on this solvent in the same manner that the films did. Bigi and collaborators discussed this point. They mentioned that gelatin films needed concentrations of glutaraldehyde above 0.25% (*v*/*v*) to not lost their integrity in hydrophilic solvents [47]. In this study, the concentration of the covalent crosslinker was 0.03% (*v*/*v*), which is clearly under the aforementioned value. Laponite increased the swelling and reduced the loss by solubilization, indicating an excellent affinity for water. Simultaneously, it could act as a physical crosslinker, aiding in the maintenance of the films’ integrity [48]. LGL and LGL MAS showed the best performance in these responses (see Table 2).

#### 3.3.4. Occlusive Effects

Table 2 enlists the means obtained from evaluating the occlusiveness of composite films vs. blanks. The occlusion factors of the blanks were the highest (*p* < 0.05); however, the presence of clay decreased the responses for empty formulations, while, in those loaded with drugs, although not statistically significant (*p* > 0.05), this led to a slight decrease, with LGL and LGL MAS being the lowest with 8.8 mg of clay. This factor depends on the sample volume, particle size, crystallinity, lipid concentration, and type of colloidal systems [49]. The absence of laponite in the blank formulations revealed a high occlusive grade as a response to the presence of amorphous lecithin, which tend to occupy more space despite being hydrofobic [29,50]. In contrast, the presence of laponite led to (1) less compaction, as the optimal formulations showed the highest thickness, (2) the development of nanovoids during the drying process formed by the house of cards structure, and (3) films being in the glassy state at room temperature [33,51]. LGL and LGL MAS did not suffer any change in thermal behavior. As seen from their rheology studies, their Tg was ≥50 °C, which will be further discussed in the DSC section. As a result, the occlusiveness might affect the bioavailability of the drugs entrapped when administered topically, as well as the gas exchange.

#### 3.3.5. Clarity

One of the essential properties of films intended for use as wound dressings is to allow the evaluation of the healing process, for which a translucent material is preferred [52]. The light transmittance for dressings is preferably less than 10% since it could prevent UV radiance to the wound, as mentioned by Pan and Kaygusuz [53,54]. In this study, the transmittances of the optimal films (three samples evaluated, *n* = 3) were 1.68 ± 0.25% and 5.13 ± 0.59% for LGL and LGL MAS, respectively. Although the films were not 100% transparent, they were in the specification range and demonstrated a slight pass of light across them due to the presence of nanovoids by laponite, as described before. The significant difference in means was because of the presence of maltodextrin D16.5–19.5 in the formulations with drugs, which improved the film network, making it more homogeneous and avoiding the agglomerated chains [45]. This assay described the project films as a protective barrier from the environment, including radiance, that will allow the healing process to be observed.

#### 3.3.6. Bioadhesion and Post-Wetting Bioadhesion

Bioadhesion describes the interfacial forces between two materials (at least one is biological) that need to be held together for some time [55]. For wound dressing, this force needs to be enough to stay in the damaged area while, at the same time, being easy to remove. In his clinical study, Waring mentioned that commercial dressings with values of over 40 cN regarding this response caused pain and extensive damage when removed, as cells and tissues remained in the material [56]. This was later confirmed by Blacklow, who suggested values close to 10 cN for this force to make them suitable for clinic application [57]. The results for LGL were −2.8 ± 0.7 cN and −5.2 ± 0.2 cN, respectively, for each response and, for LGL MAS, the results were −4.0 ± 0.4 cN and −7.4 ± 0.2 cN (three samples evaluated, *n* = 3). The negative sign correlates with the force made by the instrument to separate the surfaces. The presence of M:AS showed a statistically significant difference (*p* < 0.05). This result was evidence that the carbohydrate increased the response by its hydrophilic nature, bonding to water remnants in the skin [58]. The values in the post-wetting bioadhesion assay were close to the specifications due to the water-induced swelling of the polymer chains and clay platelets, which consequently increased the interface interactions between both surfaces. This phenomenon has a positive effect in regard to its application as a wound dressing since it is necessary to clean and wet the injured area prior to application. Thus, this will increase the force between film and skin, enabling the dressing to hold together while making it less difficult to remove.

#### 3.3.7. ATR-FT-IR

The width, intensity, and position of FTIR spectral bands are all sensitive to chemical function changes and macromolecule conformations [59]. Figure 5 shows the excipients and optimal formulations spectra. For LGL, a band was observed to be around 3300 cm^−1^ of N-H vibration of amide I of gelatin type B; however, on the composite with loaded with drugs, this band was seen as slightly wider due to O-H groups of ascorbate and maltodextrin. The bands at 2926 and 2856 cm^−1^ on the optimal formulation corresponded to C-H bonds; their intensity demonstrated the presence of lecithin as a major excipient, while the band at 1638 cm^−1^ represented the crosslinking of gelatin (-N=C-, aldimine group) [34], which was wider on LGL MAS because of the existence of carbonyl groups in ascorbate. The band at 1629 cm^−1^ confirmed the presence of amide groups of gelatin. The PO_2_ groups of phospholipids were presented by the band at 1061 cm^−1^. The appearance of a band at 970 cm^−1^ was associated with the stretching vibration of the Si–O–Si bond on laponite [11,60].

#### 3.3.8. Differential Scanning Calorimetry (DSC)

For pharmaceutical products, stability depends on the conditions of where the products will be stored, such as temperature and humidity. Temperature values should not be above 50 °C [61]. DSC is used to describe the material stability in a certain temperature range. This experiment (as reported in Figure 6) evaluated the thermal behavior of excipients on a non-optimal (LGL 6.6) and optimal (LGL MAS) formulations of films (top), as well as the derivates of each scan (below) to ensure reliable analysis. Previously, rheology assays portrayed the thermal phenomena of nanocomposite dispersions as being at 18 to 50 °C, and this will be compared and discussed at length in this section.

Regardless of laponite concentration, both films showed behavior that is typical of partially crystalline materials, with only one glass transition (Tg, green box) being followed by an endothermal melting peak (Tm, yellow box). The change in glass transition temperatures resulted from the interactions between the clay and gelatin. Compared to Tg of protein and LGL 6.6 (30.3 °C) vs. the optimal film with 8.8 mg of laponite, Tg increased by 19 °C (49.5 °C), this phenomenon was observed in the fluid behavior (Rheological analysis section) as well as in previous research such as in the work of Varnik and coworkers [62], as they interpreted the changes of Tg of polymer chains using molecular dynamics simulations. Those strongly attractive platelets (clay) led to an increase in Tg because the motion of the polymer chains, where they were embedded, was slowed in comparison to the gelatin alone. This change depended on laponite concentration—a lower concentration produced a weaker attraction and had the opposite or no effect compared to the bulk material (where the molecules can relax faster). At the same time, the other excipients probably affected the glass transition in the final product as well. The melting temperature of gelatin, which depends on the moisture [63,64], Tm = 80.8 °C [33], was higher than LGL 6.6 and the LGL MAS films (66.8 and 74.3 °C, respectively). Those changes occurred due to the presence of laponite platelets, and the polymer suffered some changes, such as loops or shorter sequences, resulting in less stable structures than the bulk materials. On the other hand, the lecithin plot had different Tg and Tm values by being a phospholipid mix. The Tm for sodium ascorbate was 194.8 °C [65]. The bands of laponite and maltodextrin corresponded to water desorption in a dehydration process, given the fact these components are hygroscopic [10,66].

The melting enthalpies were 20.4 J/g, 0.6 J/g, 0.4 J/g and 290.2 J/g [67] for gelatin, LGL 6.6, LGL MAS, and sodium ascorbate, respectively. The ∆H melting for gelatin is related to the triple helix denaturation. Thus, the enthalpies measured in the films vs. the protein were lower because laponite interfered in the formation of the triple helix of the protein. The house of cards structure increased amorphous polymer structure via electrostatic interactions with the polymer chains, avoiding the renaturation of the triple helices in the drying process. Additionally, the presence of other polymers and covalent crosslinkers influenced these values [68].

#### 3.3.9. Surface Morphology, Scanning Electron Microscopy (SEM), and Atomic Force Microscopy (AFM)

Figure 7 shows how the roughness of the films decreased in the following order: LG > LG MAS > LGL > LGL MAS (see Figure 7i,j,k,l, respectively). This reduction is due to the homogeneous distribution of clay’s platelets on polymer chains, primarily on gelatin. This led to changes in the network, resulting in amorphous structures, which were predominant in the new materials. From the micrographs, SEM and AFM (see Figure 7a,e), LG was described as having a uniform distribution of oval holes within it. Previous research revealed that the LG nanoparticles had a convex capsular form after the drying process. This translates into a more pronounced shape once the film was formed, leaving behind voids as a result the coalescence of the particles [34]. LG MAS evidenced the effects of the presence of cumulus on other excipients, such as maltodextrin (see Figure 7b,f). This film experienced agglomeration during the drying process, as well as phase separation related to the natural incompatibility of the polymers used. As a result, the holes for this formulation were wider due to its bigger particle size (see Table 1 and Figure 4c) in comparison to those presented on LG. 

The surfaces in Figure 7g,h were wider in both laponite films (LGL MAS > LGL). This result correlates with those shown in Table 2; hence, the clay was embedded in the polymer chains, increasing the thickness of the films and limiting the formation of deep craters. Still, the existence of convex shapes was observed. As previously discussed by Valencia, the reduction in the roughness could be a consequence of biopolymers’ orientation on the support surface (laponite platelets) during the drying process [20].

#### 3.3.10. Uniformity of Content

LG MAS and LGL MAS were evaluated according to the Pharmacopeia of the Mexican United States [69]. The uniformity of doses is accepted when the active amount in no less than nine out of the ten units of doses is within the range of 85.0 to 115.0%, and no amount is outside the range of 75.0 to 125.0% of the declared quantity. The previous is assessed using the method of uniformity of content. Moreover, it is established that the coefficient of variation should not be greater than 6.0%. Table 3 lists the results for the preceding formulations. It is shown that the clay containing formulation complied with the requirement of the uniformity of dose for both APIs. On the other hand, LG MAS did not fulfill this requirement for AS.

#### 3.3.11. Drug Release

Although initially contemplated, it was not possible to compare the effect of laponite upon drug release. This was due to the non-compliance with uniformity of content of LG MAS. The degradation of sodium ascorbate, as an expected response due to its photosensitivity, is accountable for this result. Ismail and collaborators suggested reductions in the movement of molecules as a strategy to avoid this phenomenon [65]. In this case, LGL MAS was able to overcome such an effect due to the presence of laponite platelets. Their positive edges attracted ascorbate anions, consequently reducing AS motion. 

Table 4 shows the results for maltodextrin (M). Approximately 90% was released within 24 h following a first-order model (r^2^ = 0.9464). Solid forms, similar to the films presented in this paper, containing hydrophilic drugs (M) in porous matrices tend to follow this profile, in which the release of the active substance is proportional to the amount of drug remaining in its interior in such a way that the amount of drug released by a unit of time diminishes [70,71].

Figure 8 for M (blue curve) displays the accumulated fraction of drug release vs. time. The slope (k) describes a time dependent release rate, as defined by Mulye and coworkers. In the first few hours, the rate was seen as constant and fast, related to a linear correlation derived between the surface area of a material and the release medium (see Figure 7h). However, release and the rate decreased proportionally with decreasing area [72]. This assay reinforced the idea that LGL MAS was a porous film due to the presence of laponite, as described in the occlusive effect discussion. Moreover, in the first few hours, the transport mechanism was fast and adequate to apply LGL MAS as dressing, allowing maltodextrin to cross from damaged to internal tissue and bring the energy necessary to accelerate the healing process [14]. The carbohydrate release was studied only for 144 h because the signal decreased over time—when the aliquots refilled the medium. The method used for this determination was not sensitive to detect low concentrations after this point.

In Figure 8, for AS (red curve), the fraction of drugs released was approximately 60% in 264 h, as determined by a Korsmeyer–Peppas model (r^2^ = 0.9680) with an *n* value equal to 0.370 ± 0.011 (see Table 4). This fitting model provided an exponent *n*, which predicts the drug transport mechanism. For *n* values ≤ 0.5, a Fick diffusion is followed. *n* = 1.0 reflects a zero-order behavior, while, for 0.5 < *n* < 1.0, the diffusion is non-Fickian (also known as anomalous transport [71]). The *n* value on this assay revealed the AS released was guided by Fick diffusion. Drug transport mechanisms were also fitted to a Higuchi model, which considers the fraction of drugs released as being proportional to t^1/2^. For this work, the model supported by Fick diffusion is considered as the main mechanism driving the displacement of drugs while embedded on a film (see Table 4). Saez found that in systems that follow Higuchi’s model, the bioactive compound was uniformly distributed on solid polymer support and could be dissolved in the polymeric matrix, as well as dispersed, if its content exceeds the solubility limit [73]. Fick’s second law is interpreted for the AS diffusion phenomenon as follows: the drug migration to the medium occurred by molecular diffusion through the support or by diffusion through existing micropores in the hybrid matrix. A decrease in the drug migration rate across time was observed as well, mainly in relation to continuous decreases in the diffusion path. This assay provided information about the efficacy of LGL MAS as a possible capable bacteriostatic dressing; hence, AS will remain in the wound to reduce pH-avoiding bacterial growth.

## 4. Conclusions

The laponite composites containing biopolymers (lecithin/gelatin), both empty and loaded with drugs (maltodextrin/sodium ascorbate), formed suspensions capable of forming in situ films, which yield attractive and innovative options for use as dressing materials, in which, the clay significantly improved the physicochemical properties of the particles and films, which is relevant to its targeted usage. Thus, the optimal particulate suspensions were prepared via the gelatin desolvation method, with 125:100 mg:mg of lecithin/gelatin mix and 8.8 mg of clay: empty (LGL), and others loaded with 10 mg of maltodextrin and sodium ascorbate (LGL MAS). This amount of laponite reduced the particulate size to 286.4 ± 28.7 and 391.8 ± 20.7 nm for LGL and LGL MAS, respectively. This can be attributed to its physical crosslinking effect. The PDI indicated a homogeneous population (<0.3), which was later observed by TEM. The dispersion exhibited an adequate Z potential—capable of avoiding agglomeration phenomenon. Additionally, the clay provided temperature stability at a range from 18 to 50 °C, and this was confirmed by rheology and DSC. Regarding the properties of the film, laponite enhanced elastic and plastic modulus to resist ruptures in the external environment (mechanical properties). Laponite also raised the swelling of the films up to 672.46 ± 24.26 and 289.06 ± 7.73% for each optimal formulation (LGL and LGL MAS) while helping to maintain the integrity of the films in water and allowing the gas exchange. Additionally, the clarity and bioadhesion should be enough to observe the healing process and stay in the wound, also making subsequent removal from the wound easier. As shown in the surface morphology section, how the synthetic mineral decreased the roughness has been described. Concerning drug release from LGL MAS, the maltodextrin provided fast and continuous transport—optimal for bringing energy to interior tissues and accelerating the healing process. In contrast, the release of sodium ascorbate was slow, meaning the drug remained in the local area, which would reduce the pH and therefore possibly avoid bacterial growth. The systems addressed in this study represent interesting, innovative, and promising alternatives to encapsulate drugs, with potential applications in medicine, especially with respect to wound dressings.

## Figures and Tables

**Figure 1 pharmaceutics-15-01634-f001:**
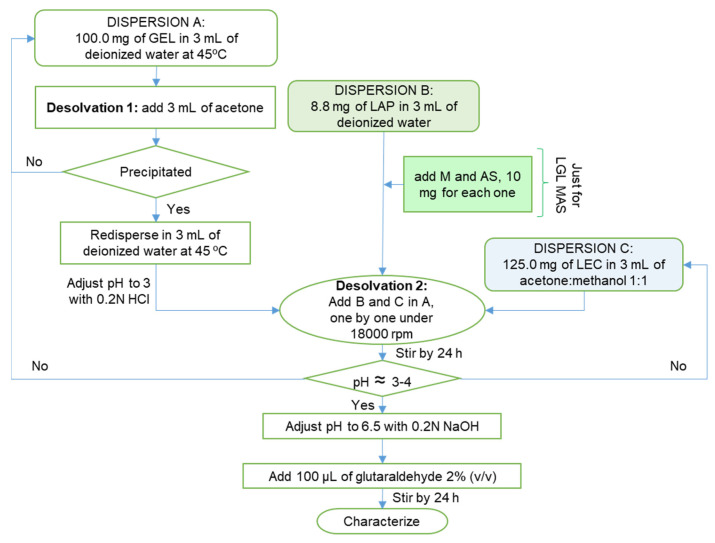
Schematic representation of the preparation of the nanoparticles.

**Figure 2 pharmaceutics-15-01634-f002:**
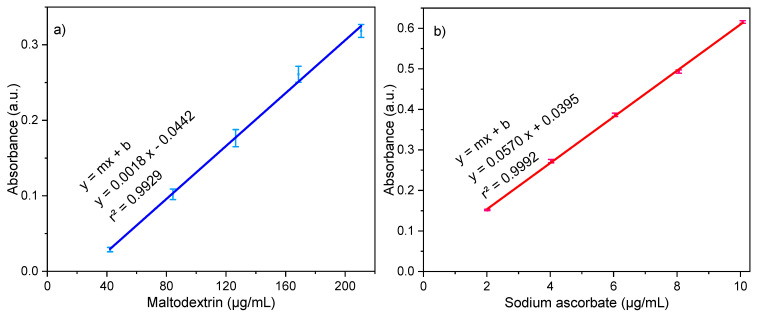
Calibration curves: (**a**) maltodextrin and (**b**) sodium ascorbate.

**Figure 3 pharmaceutics-15-01634-f003:**
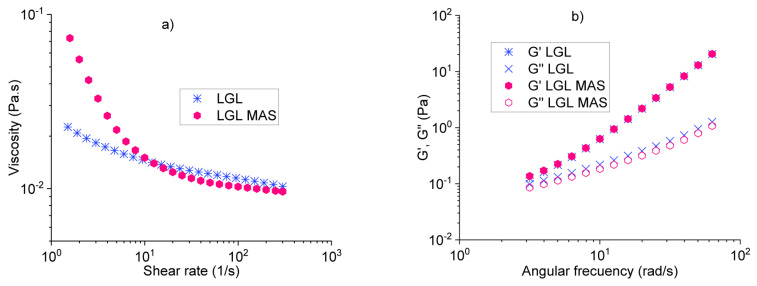
Rheology studies: (**a**) viscosity, (**b**) oscillatory study, and (**c**) thermal behavior.

**Figure 4 pharmaceutics-15-01634-f004:**
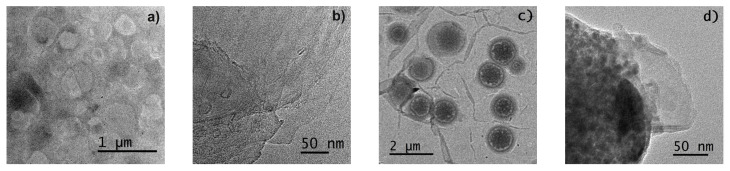
TEM of particles: LG: (**a**,**b**); LG MAS: (**c**,**d**); LGL: (**e**,**f**); LGL MAS: (**g**,**h**).

**Figure 5 pharmaceutics-15-01634-f005:**
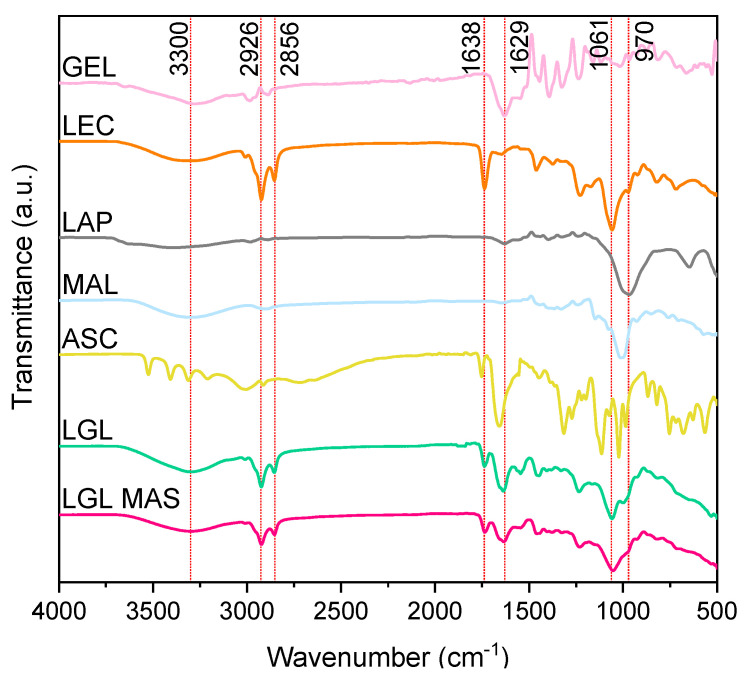
Fourier-transform infrared spectroscopy patterns for excipients and nanocomposite films.

**Figure 6 pharmaceutics-15-01634-f006:**
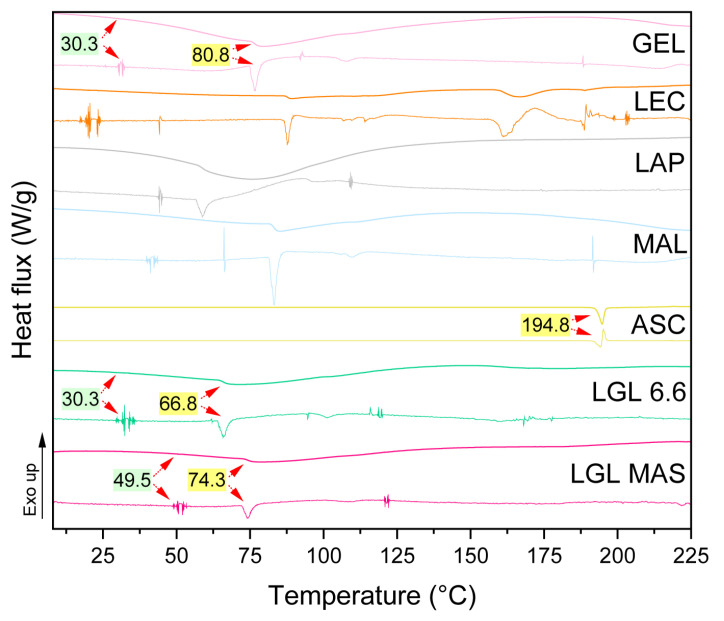
Thermal behavior determined by DSC for excipients and nanocomposite films.

**Figure 7 pharmaceutics-15-01634-f007:**
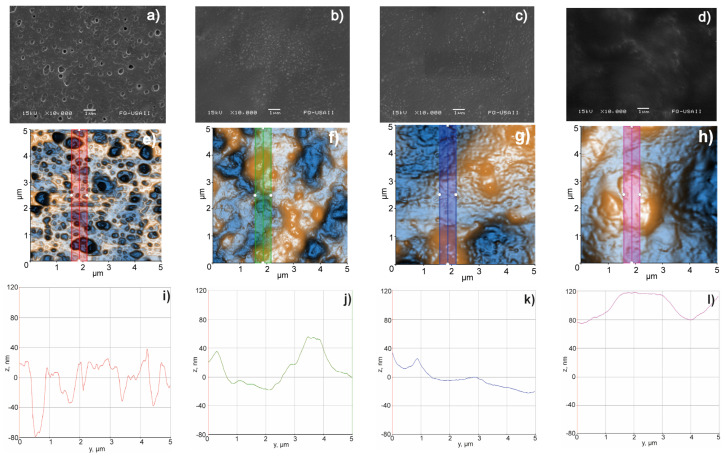
Surface morphology, SEM (**a**–**d**), AFM (**e**–**h**), and roughness (**i**–**l**) microphotograms of the following: (**a**,**e**,**i**) LG; (**b**,**f**,**j**) LG MAS; (**c**,**g**,**k**) LGL; (**d**,**h**,**l**) LGL MAS.

**Figure 8 pharmaceutics-15-01634-f008:**
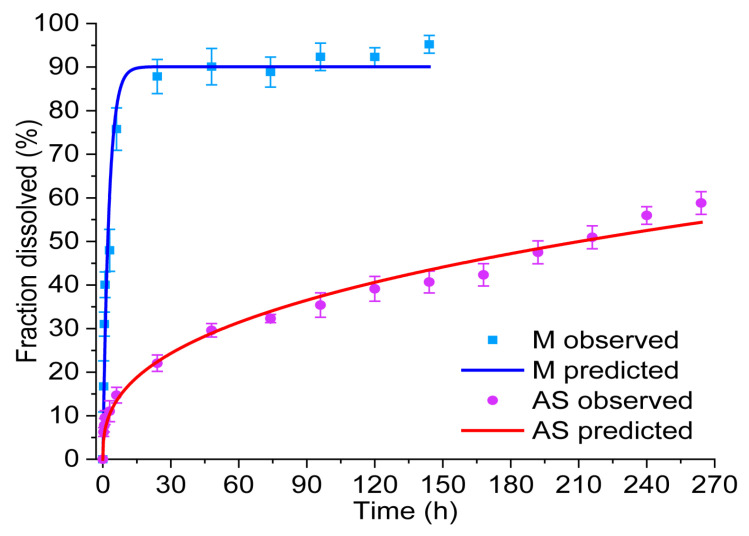
Release profile and fit of M and AS in LGL MAS.

**Table 1 pharmaceutics-15-01634-t001:** Particulate composite results and responses in terms of size, PDI, Z potential, and entrapment efficiency (*n* = 3). L (soy lecithin), G (Gelatin type B), M (Maltodextrin), and AS (Sodium Ascorbate). The factors and results with bold font describe the optimal formulation.

Sample	L:G mg:mg	Laponite mg	M:AS mg:mg	Size nm	PDI	Z Potential mV	% Entrapment	
							M	AS
LG	125:100	-	-	311.5 ± 36.4	0.179 ± 0.034	−43.2 ± 2.8	-	-
LG MAS			10:10	1801.5 ± 171.3	0.578 ± 0.183	−27.3 ± 0.8	50.03 ± 0.99	92.98 ± 0.90
LGL 4.4		4.4	-	371.8 ± 34.0	0.464 ± 0.046	−27.1 ± 1.0	-	-
LGL 4.4 MAS			10:10	1488.7 ± 118.5	0.513 ± 0.108	−15.8 ± 1.6	53.85 ± 0.98	96.59 ± 0.33
LGL 6.6		6.6	-	336.7 ± 18.9	0.625 ± 0.031	−28.2 ± 1.1	-	-
LGL 6.6 MAS			10:10	403.4 ± 65.1	0.433 ± 0.130	−16.4 ± 1.9	55.41 ± 0.25	97.68 ± 0.19
**LGL**		**8.8**	**-**	**286.4 ± 28.7**	**0.222 ± 0.008**	**−34.5 ± 1.7**	**-**	**-**
**LGL MAS**			**10:10**	**391.8 ± 20.7**	**0.145 ± 0.043**	**−24.2 ± 2.2**	**57.12 ± 0.50**	**97.96 ± 0.06**
	150:100	-	-	217.6 ± 12.3	0.295 ± 0.071	−38.2 ± 1.2	-	-
			10:10	1363.8 ± 24.2	0.508 ± 0.112	−24.3 ± 1.2	50.12 ± 0.28	93.28 ± 0.25
		4.4	-	266.6 ± 30.2	0.346 ± 0.030	−9.3 ± 0.3	-	-
			10:10	1574.3 ± 160.6	0.926 ± 0.092	−1.8 ± 0.8	54.26 ± 1.10	97.92 ± 0.50
		6.6	-	259.8 ± 4.4	0.400 ± 0.020	−12.2 ± 0.6	-	-
			10:10	772.6 ± 54.4	0.699 ± 0.044	−1.9 ± 0.3	54.81 ± 0.48	98.47 ± 0.50
		8.8	-	259.7 ± 26.2	0.408 ± 0.019	−15.1 ± 1.5	-	-
			10:10	372.6 ± 22.3	0.493 ± 0.050	−2.4 ± 1.4	57.08 ± 0.11	98.00 ± 0.39

**Table 2 pharmaceutics-15-01634-t002:** Film results, three samples analyzed for each test (*n* = 3).

Sample	Thickness mm	Young’s Modulus MPa	Tensile Strength MPa	Swelling %	Loss by Solubilization %	Occlusive Effects %
LG	0.096 ± 0.007	23.9 ± 7.2	1.8 ± 0.3	-	-	88.58 ± 1.27
LG MAS	0.101 ± 0.005	164.3 ± 26.6	2.4 ± 0.2	-	-	86.43 ± 0.48
LGL 4.4	0.097 ± 0.005	27.6 ± 6.6	0.4 ± 0.2	442.87 ± 30.65	14.55 ± 2.57	77.23 ± 3.32
LGL 4.4 MAS	0.102 ± 0.012	101.7 ± 20.3	1.6 ± 0.8	171.66 ± 21.20	19.17 ± 0.39	80.73 ± 1.12
LGL 6.6	0.102 ± 0.007	18.7 ± 3.8	0.9 ± 0.1	523.74 ± 43.61	8.28 ± 1.53	70.33 ± 1.98
LGL 6.6 MAS	0.103 ± 0.010	72.7 ± 9.8	2.9 ± 0.5	277.17 ± 40.29	9.39 ± 0.64	81.11 ± 0.68
**LGL**	**0.111 ± 0.009**	**4.7 ± 0.5**	**4.7 ± 0.8**	**672.46 ± 24.26**	**6.84 ± 0.87**	**69.56 ± 3.40**
**LGL MAS**	**0.113 ± 0.007**	**12.5 ± 2.7**	**7.1 ± 0.6**	**289.06 ± 7.73**	**6.85 ± 0.92**	**79.37 ± 1.00**

**Table 3 pharmaceutics-15-01634-t003:** Uniformity of content of LG MAS and LGL MAS.

		Drug (%)		
	LG MAS		LGL MAS	
Assay	M	AS	M	AS
1	94.07	54.23	98.53	87.19
2	98.01	45.12	97.27	91.27
3	101.28	42.73	98.83	90.01
4	95.33	55.04	97.97	85.55
5	95.58	51.09	93.74	90.88
6	94.67	55.37	98.35	86.85
7	95.99	51.56	100.43	86.91
8	92.53	51.09	96.09	86.76
9	97.19	45.31	92.70	90.25
10	101.25	47.24	100.90	85.74
Mean	96.59	49.88	97.48	88.14
SD	2.90	4.51	2.65	2.20
CV	3.00	9.04	2.72	2.50

**Table 4 pharmaceutics-15-01634-t004:** Dissolution model parameters for LGL MAS.

Drug	Parameters	Zero-Order F = kt	First-Order F = 100[1-e^kt^]	Higuchi F = kt^0.5^	Korsmeyer–Peppas F = kt^n^
M	r^2^	0.6930	**0.9464**	0.371	0.8612
	k	0.897 ± 0.070	**0.376 ± 0.026**	10.100 ± 0.472	41.139 ± 1.925
	n	-	**-**	-	0.183 ± 0.011
AS	r^2^	0.9219	0.8848	0.9501	**0.9680**
	k	0.256 ± 0.007	0.013 ± 0.001	3.575 ± 0.042	**6.896 ± 0.402**
	n	-	-	-	**0.370 ± 0.011**

## Data Availability

New data were created.

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
