# Peer review of "Laponite Composites: In Situ Films Forming as a Possible Healing Agent"

_pharmaceutics, 2023, doi:10.3390/pharmaceutics15061634_

Round 1

Reviewer 1 Report

The manuscript submitted by Pineda-Álvarez et al. reports the obtention of laponite composites with the content of biopolymers (lecithin and gelatin) for the maltodextrin: sodium ascorbate inclusion forming in situ films. The materials were characterized using techniques such as DLS, Z-Potential, FT-IR, DSC, and several microscopical techniques. Besides, rheologic experiments and evaluation of the mechanical properties were also made. The drug release using the Franz cells device was evaluated and then the obtained data was adjusted to evaluate the correlation with different kinetic models. The manuscript is well organized, and clear, and presents interesting scientific soundness. However, I am attaching some comments and suggestions to improve the research.

1-                    XRD could be an interesting technique to evaluate the laponite-based nanoparticles inside the film. 

2-                    Did the authors use any saline solution to control the ionic strength during Z-potential experiments?

3-                    I suggest the inclusion of the obtained UV-Visible data for the determination of %EE (calibration curve).

4-                    Regarding TEM images, why is the reason for coloration in the micrographs? 

5-                    Figure 3. Please, add a.u. in the y-axis and change “wavelength” to “wavenumber” in the x-axis.

6-                    Figure 5. For easy interpretation, I suggest dividing the roughness profile (5i) data in each one separately.

7-                    Figure 6. Why are release profiles measured at different times? Is not clear...

8-                    Considering the references in the manuscript are 79 in total, only 16% is literature published from 2019 to the date (last five years). I would like to ask the authors to make a little effort to update the literature, improving in this form the quality of the manuscript.

Reviewer 2 Report

The manuscript pharmaceutics-2380978 evaluates the performance of laponite in lecithin:gelatin composites (LGL) and in lecithin-gelatin composites incorporated with maltodextrin-sodium ascorbate mixture, with intended to be use as wound dressing materials.

In my opinion, the paper still needs improvements before to be published in the Pharmaceutics journal.

Abstract

- L. 15: “not toxic wakens interest”? Maybe the authors want to say “have sparked interest” or “have aroused interest”! Please revise the sentence!

- L. 14-18: How can laponite be a solution to solve the "absence of a healing material"? Maybe the authors want to say that the laponite can contribute to solving the problem. Please revise the sentence!

- L. 23: What does "pharmaceutical properties of films" mean?

- Please correct the entire presentation of the properties and structure them, so that the readers can understand them!

1. Introduction

- L. 39-50: This paragraph is not related to the manuscript and thus, must be completely removed!

- L. 51-65: There are no studies related to the wound healing in this manuscript, so the paragraph must be drastically decreased!

- L. 104: "this project can bring new composites"? How can a study bring a composite! Please be careful with for grammar and language!

- L. 104-106: “On the other hand, this project can bring new composites with the ideal conditions of healing material. Moreover, producing those substances in the lab, regarding materials and instruments, is easy and low-cost for future assays”? These are unclear assumptions, which are not demonstrated in this study! Please delete them!

- The only example referring to a composite based on laponite and alginate is presented in paragraph L. 82-87!

- The authors must bring additional rigorous information that presents the current state of composites based on lecithin, gelatin and information on maltodextrin and sodium ascorbate!

- In addition, the authors did not present the novelty of this study and did not bring clear arguments to demonstrate this fact in relation to the data from the literature.

- L. 228: “infrared” and not “infra- red”!

- L. 372: "the optimal formulations (pink and violet)"? The authors must mention the name of the samples and not the colors with which they are presented in the graph!

- L. 386: “b, d, f and h correspond to zooming of the edge of each type of particle”??

- L. 405: “Table 2. Film results (n=3)”??

- L. 427-432: If the authors discuss about swelling in wound fluid (exudates), then swelling studies must be done in different types of body fluids and not in water!

- L. 391: “the polymers were like tiny bubbles”?? Maybe particles?

As a conclusion, there are serious difficulties in the interpretation of data and some of the foundations are unsafe. Overall, there are a lot of unclear assumptions and the structure and content of the paper still needs much work! The manuscript is ambiguous and has a limited scientific merit.

The quality of the English language is low! The authors use unclear expressions, words that have a different meaning and in conclusion, the entire manuscript must be reviewed by a native English speaker!

Reviewer 3 Report

This work entitled “Laponite Composites: In Situ Films Forming as a Possible Healing Agentis an interesting article, very well written and understood. The authors have chosen up to date references. Some minor mistakes in English language can be corrected.

 I recommend this manuscript for publication in Pharmaceutics after some minor revision:

 A Schematic representation of the preparation of the LGL and LGL MAS composites would enhance the quality of the content and provide easier understanding of the materials’ preparation procedure.

Some minor mistakes in English language can be corrected.

Round 2

Reviewer 2 Report

The authors paid attention to comments and made some improvements to the manuscript pharmaceutics-2380978, as compare to previous version.

However, there are still some corrections which must be done:

-     It is not necessary to present experimental data in Abstract, but only to summarize the article's main findings! Please move the experimental data to the Conclusions section, where you can explain in detail the information obtained in this study!

-     To demonstrate that these films can be used as wound dressing, it is necessary to confirm that they are not toxic to the body and do not have a cytotoxic effect on the proliferation of human cells! In this regard, please add a new section, where to present the experimental data obtained for the biocompatibility (cell viability assay) of unloaded (LGL) and drug-loaded (LGL MAS) films!

-     L. 708: “adequate Z potential” not “adequate z potential”! 

-          The quality of English language still needs to be improved!

Round 3

Reviewer 2 Report

The authors have improved the manuscript pharmaceutics-2380978, compared to the final version and thus, I agree to its publication in its current form.

The way the manuscript was formulated in its initial form (Introduction section), insisting a lot on wound management, cell proliferation and the ideal conditions for accelerated healing, imposed to the authors to provide more information regarding the biocompatibility and cytotoxicity of the obtained material. But, in the current form it is clearly understood that the performance of the films is evaluated, with an emphasis on the presentation of their physicochemical properties.

Minor editing of English language required